# Are There Definite Disease Subsets in Polymyalgia Rheumatica? Suggestions from a Narrative Review

**DOI:** 10.3390/healthcare13111226

**Published:** 2025-05-23

**Authors:** Paolo Falsetti, Ciro Manzo, Marco Isetta, Francesco Placido, Alberto Castagna, Maria Natale, Caterina Baldi, Edoardo Conticini, Bruno Frediani

**Affiliations:** 1Rheumatology Unit, Department of Medical Sciences, Surgery and Neurosciences, University of Siena, 53100 Siena, Italy; f.placido.aif@gmail.com (F.P.); catebaldi3@gmail.com (C.B.); conticini.edoardo@gmail.com (E.C.); fredianibruno60@gmail.com (B.F.); 2Rheumatology Outpatient Clinic, Internal and Geriatric Medicine Department, Azienda Sanitaria Locale Napoli 3 Sud, 80065 Sant’Agnello, Italy; manzoreumatologo@libero.it (C.M.); maria.natale.int@alice.it (M.N.); 3Central and North West London NHS Foundation Trust—Clinical Education, London UB8 3NN, UK; marcoisetta192@gmail.com; 4Department of Primary Care, Health District of Soverato, Azienda Sanitaria Provinciale Catanzaro, 88100 Catanzaro, Italy; albertocastagna78@gmail.com

**Keywords:** polymyalgia rheumatica, subset, giant cell arteritis, calcium pyrophosphate deposition disease, chondrocalcinosis, immune checkpoint inhibitor drugs, infection, vaccination, ultrasound

## Abstract

**Background:** Polymyalgia rheumatica (PMR) has a multifaceted onset and course, and making a distinction between *true* PMR and so-called “polymyalgic syndrome” (that is, similar manifestations caused by different conditions) is far from easy in clinical practice. The existence of subsets within *true* PMR may further complicate the diagnostic question. Distinguishing PMR subsets from PMR-mimicking conditions does not just carry nomenclature value and speculative significance. Indeed, the correct diagnosis influences treatment, prognosis, epidemiological assessments, and health policies. **Objectives**: We aimed to (1) ascertain the presence of a definite and peculiar subset/subgroup/cluster of PMR in the scientific literature; (2) describe any possible subset/cluster/subgroup of PMR identified in at least two different studies. **Methods:** We performed a non-systematic (PRISMA protocol not followed) literature search on Embase and Medline (OVID interface). The following search terms were used: polymyalgia rheumatica, subset, cluster, subgroup, subclinical giant cell arteritis, mimicking conditions, polymyalgia rheumatica-like conditions, immunotherapy, checkpoint inhibitor, acute-phase reactants or acute-phase proteins, vaccination, infection, and calcium pyrophosphate deposition disease or chondrocalcinosis. Each paper’s reference list was scanned for additional publications meeting this study’s aim. Abstracts submitted at conferences or from non-peer-reviewed sources were not included. **Results**: The initial search yielded 2492 papers, of which 2389 articles were excluded based on title and abstract screening. A total of 103 articles underwent a full-length review, and 84 of them were finally assessed for eligibility. A total of seven large subsets of PMR could be identified: (1) PMR with normal acute-phase reactants; (2) PMR with an infection trigger; (3) PMR with a vaccination trigger; (4) PMR with subclinical giant cell arteritis (GCA); (5) PMR and calcium pyrophosphate deposition disease (CPPD); (6) PMR following immune checkpoint inhibitor (ICI) therapy; (7) PMR with peculiar clinical clusters (based on clinical or statistic clustering methods). **Conclusions**: PMR with normal baseline acute-phase reactants and PMR with an infection or a vaccination trigger could be categorized as subsets of disease. PMR with subclinical GCA and most cases of PMR/CPPD should be categorized as mimickers. Finally, further studies are required to better categorize some peculiar clinical subsets emerging from cluster analyses, and ICI-induced PMR.

## 1. Introduction

Polymyalgia rheumatica (PMR) is a common condition in the elderly, characterized by inflammation of the shoulders, neck, and pelvic girdle, often associated with systemic manifestations such as low-grade fever, weight and appetite loss, general malaise, and sleep disorders [1]. Diagnosis is clinical, but imaging tools like magnetic resonance imaging (MRI) and ^18^F-labeled fluorodeoxyglucose-positron emission tomography-computed tomography (^18^F-FDG PET-CT) have recently gained growing importance in characterizing this condition [2,3], whereas ultrasonography (US) has been included in 2012 ACR/EULAR provisional classification criteria [4]. These criteria were designed to discriminate between patients with PMR and those with PMR- mimicking conditions and are not meant for diagnostic purposes. To date, other imaging tools are still awaiting inclusion in validated diagnostic or classification criteria. On the other hand, some diagnostic criteria have been proposed, with the highest sensitivity (89%) for Bird et al.’s criteria [5]. Although there are validated diagnostic and classification criteria some investigators use local protocols.

At present, no laboratory test ha specificity for PMR diagnosis. Acute-phase reactants (APRs) such as erythrocyte sedimentation rate (ESR) and C-reactive protein (CRP) are usually raised at the onset of disease, but the diagnosis of PMR has been described even with normal ESR and CRP [6,7,8]. Alternative biomarkers—such as plasma fibrinogen or interleukin-6 (IL-6) serum concentrations—have been proposed. However, their usefulness and feasibility in everyday clinical practice are still awaiting confirmation in large case series [8]. Moreover, no definite clinical, laboratory or imaging findings can predict prognosis, different clinical courses of PMR [9,10] or the risk of disease relapse [11,12,13].

Therapy relies on glucocorticoids (GCs), which usually lead to a rapid reduction in both pain and stiffness, with progressive normalization of APRs. Nevertheless, PMR is associated with significant morbidity related to long-term GC side effects [14]. Moreover, relapses occur in up to half of PMR patients in GC standard therapy [12,13].

Because of the lack of specific diagnostic tests, PMR diagnosis requires the exclusion of other conditions with similar presentations, commonly defined as PMR-like conditions or PMR-mimicking diseases. Differential diagnosis is far from easy. Some patients with PMR-mimicking conditions can have a fast but transitory response to systemic GC. Changes in final diagnosis have been experienced in about half of patients with initial manifestations of PMR; in most cases, their diagnosis has been changed to chronic arthritis [15,16,17,18,19]. In at least 20% of patients, PMR can be associated with subclinical giant cell arteritis (GCA) [20,21,22], at onset or in the long-term. Moreover, several studies have reported an association of PMR with previous infections, vaccinations, or cancer treatment with immune checkpoint inhibitor (ICI) drugs, where differences with idiopathic PMR are anything but clear [23,24,25]. In addition, some PMR patients can experience a late diagnostic shift when subject to in-depth imaging [26,27,28].

Because of the disease’s multifaceted onset and course, making a distinction between *true* PMR and so-called “Polymyalgic Syndrome” (that is, similar manifestations caused by different diseases/conditions/disorders) is far from easy in clinical practice [18,19,29,30]. The existence of subsets within true PMR may further complicate the diagnostic question. In other words, does the patient suffer from a subset of true PMR or a PMR-mimicking disease?

Definitions of what a subset (or subgroup or cluster) of true PMR is and what a PMR-mimicking condition is are proposed by the authors after a discussion is held and a consensus is reached:(a)**Subset or subgroup or cluster of true PMR:** Patients with diagnosis of a PMR, fulfilling a set of diagnostic or classification criteria, and therefore having peculiar clinical and/or laboratory and/or imaging and/or outcome findings. The possibility that diagnosis was based on local protocols rather than on validated criteria was also accepted. Finally, the peculiar characteristics could also be defined by statistical methods (i.e., cluster analyses).(b)**PMR-mimicking conditions:** _Patients initially treated as having PMR who fulfil a validated set of criteria for another nosological entity (illness, disease) within a short or long follow-up.

Given this background, we performed a narrative review with the following as our primary objectives:(1)To ascertain the presence of a definite and peculiar subset/subgroup/cluster of PMR using wider case study of pure PMR diagnosed in accordance with clinical, diagnostic or classification criteria;(2)To describe any possible subset/cluster/subgroup of PMR identified in at least two different studies.

## 2. Materials and Methods

### 2.1. Search Strategy

On 2 December 2024, one of the authors (Isetta, M) performed a non-systematic (PRISMA protocol not followed) and comprehensive literature search on Embase and Medline (OVID interface). The following search terms were used: polymyalgia rheumatica AND subset OR cluster OR subgroup, subclinical giant cell arteritis, mimicking conditions, polymyalgia rheumatica-like conditions, immunotherapy, checkpoint inhibitor, acute-phase reactants, acute-phase proteins, vaccination, infection, calcium pyrophosphate deposition disease, and chondrocalcinosis—both MESH headings and free texts were searched. Searches were performed regardless of language and time of publication. Our review had no registration number.

Abstracts submitted at conferences and non-peer-reviewed papers were excluded. Additionally, the reference list for each of the selected articles was carefully read to identify any other articles of interest.

### 2.2. Data Extraction

A single author (Isetta, M) screened all titles of the identified articles against the above criteria, and subsequently, two of the authors (Falsetti, P and Manzo, C) independently screened their abstracts. After this step, data comparisons were conducted to ensure completeness and reliability, and reasons for exclusion were recorded. Where present, differences in opinion were discussed by all authors and resolved by consensus.

Finally, the full texts of all potentially relevant articles were sourced. Specifically, we considered all studies and case reports describing any subset, subgroup or cluster of PMR, and in which this subset/subgroup/cluster was compared with typical PMR (possibly within the same study).

## 3. Results

### 3.1. Description of Included Studies

The initial search yielded 2492 papers, of which 2389 articles were excluded based on title and abstract screening. A total of 103 articles underwent a full-length review, and 84 of them were finally assessed for eligibility.

A total of seven large groups of patients with PMR findings could be identified: (1) PMR patients with normal baseline acute-phase reactants (APR); (2) PMR patients with an infection trigger; (3) patients with PMR following vaccination; (4) PMR patients with subclinical giant cell arteritis (subGCA); (5) PMR and calcium pyrophosphate deposition (CPPD) disease patients; (6) patients with PMR following immune checkpoint inhibitor (ICI) therapy; (7) PMR patients with peculiar clinical subsets (possibly based on statistic clustering methods).

We resume each subset in both a different paragraph and table.

### 3.2. PMR with Normal Baseline Acute-Phase Reactants (APRs)

Three retrospective studies extensively assessed the characteristics of PMR patients without elevated baseline ESR and CRP (Table 1) [6,7,31]. In addition, isolated reports were present in two other articles. Specifically, Norwegian clinicians found both normal ESR and normal CRP in 1.2% of 178 PMR patients [32]. Additionally, only 1 patient amongst 177 had normal ESR and normal CRP in a prospective follow-up study conducted in two Italian secondary referral centers of rheumatology [33]. No alternative diagnosis to PMR was possible in all these reports.

Significantly different percentages of PMR patients with normal baseline ESR and CRP concentrations were reported in the studies listed in Table 1. Specifically, Manzo et al. [7] reported a very low percentage (1.52%) of patients compared to the percentages reported by Marsman et al. (13.6%) [6] and by Kara et al. (14.8%) [31]. Differences in inclusion and exclusion criteria, and very different follow-up times, could explain these differences. However, all three studies agreed on the need to utilize imaging (US assessment, primarily) as well as measurement of other biomarkers in all patients who have a clinical suspicion of PMR but not raised ESR and CRP, as already proposed in a 2018 editorial article [8]. Moreover, taking together all the data from these three studies, PMR with normal baseline APR should be categorized as a subset with an atypical presentation, milder systemic manifestations and longer average times for correct diagnosis. Specifically, the investigators considered milder systemic symptoms the consequence of a failure to increase IL-6 levels (with the result of normal baseline values of CRP and ESR, and an absence of anemia) [34]. Additionally, normal baseline values of ESR and CRP could in themselves justify the longer average times in correct diagnosis. Noticeably, no case of GCA was diagnosed during follow-ups.

### 3.3. PMR with Infection Trigger

Several infectious agents have been held responsible for PMR over time. Recently, reports on cases of PMR following COVID-19 disease have revived the role of infection as an etiological or triggering factor. However, no clear-cut association has yet been identified.

The possibility that PMR following infective triggers may be a different subset of disease has been discussed in the published literature (Table 2) [23,35,36]. Specifically, in a 2020 Italian retrospective study, three patients reported upper respiratory tract infection, five reported seasonal influenza, and one reported lower respiratory tract infection (pneumonia) before the onset of PMR. A correlation between infective triggers and higher CRP at diagnosis, faster response to therapy, and milder shoulder synovitis was found in these patients. No cases of GCA were identified during follow-up. According to the authors of this study, PMR triggered by infection could constitute a subset of disease. To the best of our knowledge, this is the only study reporting the possibility that PMR following infections can be considered a subset of disease [23].

Another Italian observational study was carried out on cases of inflammatory rheumatic diseases (IRDs) with an onset after SARS-CoV-2 infection or COVID-19 vaccine administration. PMR was diagnosed in 28/122 patients (22.9%) in the post-SARS-CoV-2 cohort. Of these 28 PMR patients, 1 was below 50 years of age. Concurrent GCA was excluded based on clinical features. GCs were effective in 100% of PMR patients. The data provided by the authors of this study, however, did not allow them to determine whether or not PMR following SARS-CoV-2 infection was a subset of disease [36].

More recently, a narrative review concluded that all data available in the published literature on the possible existence of a subset of PMR following infections are poorly assessable [37].

### 3.4. PMR Following Vaccination

Our literature search retrieved many case reports or case series of PMR following vaccination. Recently, reports based on pharmacovigilance databases developed after the COVID-19 pandemic and subsequent vaccination campaigns have been published. However, pharmacovigilance databases offer generic data about long-term outcomes.

In all the articles listed in Table 3 [23,36,38,39,40,41,42,43,44,45,46,47,48,49,50,51,52], no changes in diagnosis or suggestions for mimicking conditions are reported. Generally speaking, scant data are available on the characteristics of PMR patients with post-vaccine onset with respect to idiopathic cases, and no significant differences in both presentation and outcome can be discerned among that different types of vaccines causing PMR. Many of the included articles did not show any clinical, definite subset of post-vaccinal-onset PMR [41,42,43,45,46,49]. Nevertheless, post-vaccine onset of PMR was more frequently described in females [45,47]; one study highlighted a mean age slightly inferior to that for idiopathic PMR [36], and French investigators reported on a self-limiting course in post-influenza-vaccination PMR [40].

On the other hand, some studies suggested that post-vaccine PMR could be a subset of disease, characterized by better outcomes, with a shorter course of the disease [23,47,51], lower relapse rates [47,51], and lower GC cumulative dosages when compared to idiopathic PMR [48,51]. In addition, a prevalent, inflammatory involvement of the pelvic girdle on imaging was reported in two of these studies [23,48].

Lastly, all the studies listed in Table 3 include no suggestions for mimicking conditions, except for very few studies in which autoimmune/inflammatory syndrome induced by adjuvants (ASIA) is suspected [38,39].

### 3.5. PMR with Subclinical GCA (subGCA)

PMR with subGCA at onset has been described in several studies, with different modalities of diagnosing GCA [20,21,22]. In the last decade of the previous century, temporal artery biopsy (TAB) and clinical diagnosis of GCA were prevalent, whereas in recent years, PET/CT and CDUS have been the most commonly used technologies for the diagnosis of subGCA. Recently, classification criteria proposed for GCA [53] include vascular CDUS, among imaging criteria. The technologic improvements in imaging diagnostic tools included increased sensitivity and consequently produced a progressive increase in cases in which subGCA was recognized. In fact, subGCA frequency in patients with PMR rose from 2–8% in the oldest studies to 20–27% in a more recent meta-analysis. Moreover, these percentages increased to up to 66% when an in-depth imaging study (PET/CT) was applied to PMR that was resistant to therapy or relapsing [22,54,55,56].

The majority of the studies listed in Table 4 agreed on a more severe course and prognosis of PMR with subGCA, finally requiring more aggressive and/or prolonged therapy.

An older age at the time of PMR diagnosis was more frequently reported in the sub GCA group in various studies [57,58,59,60,61] except one [21]. Similarly, the majority of studies reported a significant difference in genders in patients with subGCA, with a prevalence of females [54,59,62,63,64,65]. Characteristically, all patients with PMR and late-onset GCA were females in one study [63].

Higher levels of ESR and/or CRP concentrations at onset [21,55,57,62,65,66,67,68], thrombocytosis [62,64], lower hemoglobin [62,64] and a more severe course [20,58,62,63,69] were common findings in many studies.

Differences in clinical presentation were reported in only a few studies. In particular, a recurrent clinical characteristic of PMR with subGCA could be a higher frequency of inflammatory low back pain or pelvic girdle inflammatory pain [21,60,61,64,66].

In the conclusions of various studies, the authors did not make a uniform judgement on the question of if PMR with subGCA could be a more severe subset of PMR disease in the spectrum of PMR/GCA diseases, or a more definite different diagnosis of LVV. However, PMR patients with subGCA usually had a more severe and relapsing course, so they required more aggressive and/or prolonged therapy in almost all of the studies [20,21,58,59,62,63,69].
healthcare-13-01226-t004_Table 4Table 4Polymyalgia rheumatica with subclinical giant cell arteritis.ReferenceStudy DesignStudy Sample (Peculiar PMR Patients/Total Sample)Diagnosis of PMRLength of Follow-UpImagingPresence of Definition of Subset/Subgroup/ClusterSignificant Characteristics of Subset/Subgroup/ClusterSuggested PMR-like Condition/Other Nosologic EntityGonzalez-Gay et al. [62]monocentricretrospective45 PMR with GCA in TABvs. 117 pure PMRACR criteria for GCA
TAByes: PMR with subclinical GCA in TABpMR with subclinical GCA: predominantly women, longer disease duration, higher inflammation, PLT, constitutional symptoms, lower Hb, more severe courseno, but different prognosisBlockmans et al. [54] monocentric retrospective69 PMR25 GCA or PMR12 TAB-PMRHunder and Healey criteria2 y^18^F-FDG PET-CT and TABPMR with subclinical LVVpredominantly females, no differences in inflammatory markers and age,yes, PMR as an LVV Schmidt et al. [57]monocentric prospective102 pure active PMR8% GCABird and 1990 ACR GCA criterianaCDUS e TABPMR with subclinical GCA on CDUSolder, higher ESRdifferent diagnosis and treatmentGonzalez-Gay et al. [58] 
89 severe PMR 8 (9%) subclinical GCA TAB2% in overall PMR na2 yTAByes: severe PMR ESR > 80, constitutional symptomsolder, more severe courseprobably different conditions, different therapies (GC dose) and different coursesCantini et al. [69] monocentricretrospective76 pure PMR12/76 subGCAHealey criteria6 yTAByes: subclinical GCA TABmore severe course and cranial symptomsno definitive conclusions, common genetic backgroundCimmino et al. [55]case series8 steroid-resistant PMR3/8 LV-GCAna64 months ±61.4^18^F-FDG PET-CT and TABsubclinical GCA: FDG uptake ≥2 in any vesselsubclinical LV-GCA: more frequent in females, higher CRP (146 vs. 44) and ESR (103 vs. 65)possible different diagnosis, suggestion to treat with steroid-sparing drugs as GCANarvaez et al. [63] monocentric retrospective18 PMR (11%) with late GCAHealey and 1990 ACR GCA criteria3 m–4.5 y (mean 7 mo)TABPMR with late GCAall females, ischemic symptoms, more severe courseno, high-risk and not benign PMR subsetLavado-Perez et al. [56]monocentric prospective40 consecutive atypical PMR26 (65%) subclinical LVVnana^18^F-FDG PET-CTatypical PMR (lack of treatment response)no difference between group LVV and no LVVyes, diagnosis of LVVdo et al. [70]monocentricretrospective54 PMR4 subGCA (7.4%)EULAR ACR criteriana^18^F-FDG PET-CTsuclinical GCA in ^18^F-FDG PET-CTnayes, suggestion for different diagnosisLiozon et al. [59]multicentricretrospective67 PMR late GCA65 pure PMR130 pure GCAGCA: ACR 1990 criteria PMR: clinical diagnosis and follow-up38.5 months(range 3–132)^18^F-FDG PET-CT, TAB, CT or US in selected casesyes: subset of PMR with late development of GCAPMR with late GCA: (after median 17 months), more frequent in females, older, subclinical aortitis (OR 6.42), fewer headache and feveryes, possible subclinical GCA (suggestion to treat with steroid-sparing drugs as GCA has high risk for blindness)Prieto-Peña et al. [66]monocentricprospective84 classic PMR; 60.7% subGCAACR EULAR criteriana^18^F-FDG PET-CTyes: new onset PMR with subclinical LV-GCAPMR with subclinical GCA: lower limb pain (OR 8.8), pelvic girdle pain (OR 4.9), inflammatory LBP (OR 4.7)PMR and GCA as a spectrum of the same disease. No specific conclusionsvan Sleen et al. [67]monocentric prospective39 pure PMR10 PMR GCAChuang criteria46 mo (0–76)34 mo (3–69)^18^F-FDG PET-CT and TABPMR with concurrent GCA at diagnosisPMR GCA: higher ESR, angiopoietin-2no, subset of PMR with unfavorable prognosis, requiring DMARD at onsetHemmig et al. [64] review-566 new onset PMR-subGCA pooled 23% (20% TAB, 15% US, 29% PET-CT)-243 new onset PMR analyzed by IPD Individual patient data-65 (27%) subclinical GCA in PET-CTvarious criteria
^18^F-FDG PET-CTyes: new-onset PMR with subclinical GCAPMR with subclinical GCA: inflammatory back pain (OR 2.73 and no lower limb pain (OR 2.35), in females (OR 2.31), with weight loss (1.83), fever (OR 1.83) thrombocytosis (OR 1.51); reduced OR (0.80) for higher hemoglobin levelsPMR and GCA as being on aspectrum of the same disease; no specific conclusionsCamellino et al. [71]prospective84 PMR42 LVV subclinical (50%)birdna^18^F-FDG PET-CTsubclinical LVV in pure PMRno clinical predictor of subclinical LVVPMR and GCA as spectrum of the same diseaseNielsen et al. [22]systematic review and meta-analysisPMR with subGCA 6–66%point-prevalence 22%various




Colaci et al. [65]retrospective monocentric17/80ACR/EULAR criteriaat least 1 year^18^F-FDG PET/CTyes: PMR patients who underwent ^18^F-FDG-PET/CT because of a persistent increase in acute-phase reactants besides the steroid therapymore frequent in females, higher CRP and ESR, higher grades of articular and periarticular inflammation (suggesting chronic arthritis), subclinical GCA in about a third of patientsYesManzo et al. [68]retrospective single-centre143 PMR23/143 (16.1%) subGCAACR/EULAR criterianaUS and CDUSyes: patients with halo signs in at least one examined artery were considered to have subclinical GCAPMR with subclinical GCA had shorter (<45 min) morning stiffness, higher ESR and CRP.noBurg et al. [21]prospective monocentric60 PMR28/60 GCA/PMRACR/EULAR criteria and ACR 1990 classification criteria for GCA6 monthsUS and CDUSyes: patients with halo signs in at least one examined artery were considered to have subclinical GCA.PMR with subclinical GCA (GCA/PMR = 46%):younger (69 vs. 74 y), shorter disease duration (10 vs. 16 w), higher CRP (cutoff 26.5 mg/dL), lower frequencies of effusions in shoulder and hips, but higher frequencies in hips.yes: PMR patients with subclinical GCA were treated as having GCAHemmig et al. [72] retrospective49/311 GCA had prior PMR (mean 30.5 months before)ACR 1990 GCA criteria2006–2021CDUS
51% of prior PMR patients had LVV, and lower ESR and cranial symptoms44.9% received 9.5 mg GC (diagnosis PMR > 30 months before!)possible different diagnoses, suggestion to treat with steroid-sparing drugs for GCA with LVV phenotypeDe Miguel et al. [60]Prospective multicentric79/346ACR/EULAR criteria
US and CDUSYes: Patients with halo signs in at least one examined artery were considered to have subclinical GCA.PMR with subclinical GCA: older, longer morning stiffness, more frequently reported hip painNoDe Miguel et al. [20] Prospective multicentric50/150ACR/EULAR criteria2 yearsUS and CDUSYes: Patients with halo signs in at least one examined artery were considered to have subclinical GCAPMR with subclinical GCA had higher number of relapses during follow-up, especially when treated with lower starting doses of GCpossible (suggestion to treat as GCA)Moreel et al. [73]retrospective monocentric337 PMR31/337 (9%) subGCA
12 months^18^F-FDG PET-CTsubclinical GCA: FDG uptake ≥2 in any vesselPMR with subclinical GCA: higher doses GC in first 12 months, no differences in relapse rate and duration GCGPSD (suggestion for possible different outcomes)Cowley et al. [61]review and meta-analysis of two studies [20,21]107 PMR/subGCA299 pure PMRACR/EULAR and 1990 ACR GCA criteriamax 2 yyCDUS/USyes: patients with halo signs in at least one examined artery were considered to have subclinical GCAolder age at the time of PMR diagnosis and higher incidence of hip girdle symptoms were more frequently reported in the subclinical GCA grouppossible: the medium- term clinical outcome of subclinical GCA in PMR with a more severe phenotype was an increased rate of relapse and a higher GC and DMARDs;those who relapse on higher GC doses (≥10 mg) with subclinical GCA should be gconsidered for early DMARDPMR = polymyalgia rheumatica, GCA = giant cell arteritis, ACR/EULAR = European League Against Rheumatism/American College of Rheumatology, US = ultrasound, CDUS = color doppler ultrasound, GC = glucocorticoids, ESR = erythro-sedimentation rate, CRP = C-reactive protein, COVID-19 = coronavirus disease 2019. MRI = magnetic resonance imaging, ^18^F-FDG PET-CT = ^18^F-labeled fluorodeoxyglucose-positron emission tomography-computed tomography, TAB = temporal artery biopsy, US = ultrasound, GC = glucocorticoids, na = not assessed, OR = odds ratio, LVV = large-vessel vasculitis, LBP = low back pain, GPSD = GCA/PMR spectrum disease; subGCA = subclinical giant cell arteritis.

### 3.6. PMR with CPPD

The first description of CPPD in PMR patients by Dieppe [74] suggested either a chance association for the co-existence of the two pathologies or that the steroid treatment prescribed to patients with PMR features might predispose them to the development of chondrocalcinosis. All the studies reporting an association between PMR and CPPD (and/or chondrocalcinosis) are listed in Table 5 [17,18,19,74,75,76,77,78,79,80,81,82,83].

A few studies report the concurrent presence of CPPD (and/or chondrocalcinosis) in cohorts of patients with definite pure PMR [18,19,20,74,76,80,82,83]. It is noteworthy that only studies with an extensive use of imaging (particularly CR and US) described this subset, whereas studies with only clinical observations failed to demonstrate any significant presence of such association [15,84].

Several case report and case series describe so-called “crowned dens syndrome” (atlo-axial involvement of CPPD with calcification of the transverse ligament of the atlas) as a PMR mimicker [75,77,78,79,81].

All the more recent cohort studies agreed to propose a PMR/CPPD subset with recurrent characteristics with respect to pure PMR: older females, with lower levels of inflammatory markers at onset, a higher frequency of peripheral arthritis, lower exudation on the bursa sites of shoulders, and a presence of diffuse and typical deposits suggestive for chondrocalcinosis on imaging [17,18,19,76,80]. All the studies suggest that this subset usually requires shorter steroid therapy. One study also suggests a good response only to NSAID in PMR with concurrent CPPD [80]. Detection of CPPD in patients with suspected PMR seems relatively frequent, ranging between 8% and 48%. All the studies substantially agree on considering the subset PMR/CPPD as a different nosologic entity with respect to pure PMR, but diagnostic criteria for CPPD largely differ among the studies (conventional radiography or US evidence of CC, SFA, or not specified) [17,18,19,74,75,76,77,78,79,80,81,82,83].

### 3.7. PMR Following Immunotherapy with Immune Checkpoint Inhibitor (ICI) Drugs

All the studies reporting a diagnosis of PMR following immunotherapy with ICIs (ICI-PMR) and the main differences between ICI-PMR and pure, primary PMR are listed in Table 6 [25,85,86,87,88,89,90,91,92,93,94,95,96,97,98,99]. The relationship between immunotherapy with ICIs and PMR is still debated [100,101] and Three articles should be more detailed to better understand the topic [25,94,99].

In 2022, de Fremont et al. compared 14 ICI-PMR with 43 primary PMR patients and pointed out a high male-to-female ratio (6:1) and a great incidence of peripheral arthritis (28%) in the ICI-PMR cohort. In addition, about a quarter of ICI-PMR patients required DMARDs (methotrexate and tocilizumab, primarily) to effectively manage disease activity [94].

More recently, Vermeulen et al., comparing 15 patients with ICI-PMR against 37 patients with primary PMR, found other relevant differences between ICI-induced PMR and primary PMR patients [99]. Specifically, ICI-PMR patients presented the following: (1) milder clinical manifestations; (2) lower acute-phase reactant (APR) values; (3) less likely fulfilment of the 2012 EULAR/ACR classification criteria; (4) lower inflammation at ^18^F-FDG-PET/CT, (5) lower GC dosages, in comparison with those of patients with primary PMR.

Finally, based on an analysis of 38 studies, Hysa et al. identified 314 cases of ICI -PMR [25]. Interestingly, they found great variability in the PMR onset range (from one day to 53 months) after the initiation of ICI treatment, and in manifestations and outcomes. Specifically, APR values were in their normal ranges in some patients: in these patients, imaging—mainly via US and PET/CT scans—provided valuable insights for the correct final diagnosis. Many patients had a better clinical response to GCs, and yet 20% required additional DMARDs to manage their disease activity. Finally, many ICI-PMR patients experienced fewer relapses, with a rate of approximately 1.4%, which is significantly lower than the relapse rate reported in primary PMR.

### 3.8. PMR with Peculiar Clinical Subsets (From Clustering Methods or Clinical Observations)

All the studies reporting peculiar PMR subsets (derived from a clustering methodology or clinical observations) are listed in Table 7 [9,19,23,71,102,103,104,105,106,107].

According to our literature search, only three studies [9,19,107] defined peculiar outcome subsets of PMR by cluster analysis based on the continuous variables available at the onset of the disease.

Specifically, Hayashi et al. described three clusters: one with severe inflammation and a worse outcome; another with lesser systemic and articular inflammation, with persistent PMR diagnosis and only GC; and the last with an arthritic evolution [106]. In a study by Muller et al., similar patterns recurred: a cluster with a higher clinical severity and poorer outcome; a cluster of older male patients with longer-lasting stiffness and persistent PMR diagnosis; a cluster with a possible diagnostic shift [9]. Lastly, Conticini et al. described a cluster of older male patients, with typical shoulder symptoms and lower systemic and articular inflammation, who had a persistent PMR diagnosis and GC therapy that lasted longer [19]. In other words, the subset characterized by the male sex, advanced age, longer-lasting stiffness, lower systemic and articular inflammation, and typical shoulder symptoms seemed to be better associated with persistent PMR diagnosis in the long term. In the studies where detailed US and/or MRI imaging was available, this subset of PMR patients showed lower joint synovitis in the shoulder, but higher extra- and periarticular pattern of inflammation in the shoulder and hip [19,23,71,105,106,107].

On the other hand, observations of a subset of younger female patients characterized by higher inflammation, more frequent peripheral synovitis, worse responses to GC and a need to use DMARDs were recurrent in various studies [9,71,103,105,106]. These patients were frequently diagnosed late as having subclinical GCA or chronic arthritis. Consequently, these characteristics should be regarded as a warning signal for a different diagnosis.

## 4. Discussion

Our literature search highlighted significant heterogeneity in the modes of PMR diagnosis and classification. Despite the validated diagnostic or classification criteria, some researchers used local protocols (for example: [6,31,41,46]) or did not report on how the PMR diagnosis was made ([47,48,49,56,74,75] among these).

We chose to analyze all the peer-reviewed studies in which a subgroup was described, independently from the diagnostic modalities. This approach could be useful to develop an initial idea of the heterogeneity of PMR, especially in diagnostic modalities.

PMR is estimated to be one of the most common inflammatory rheumatic diseases in the elderly. Nevertheless, its recognition is not always straightforward in everyday clinical practice. To determine if patients suffer from true PMR or mimicking conditions has significant consequences: for example, in patients with PMR-mimicking conditions, it is mandatory to treat the underlying disease. The same applies to prognosis. On the other hand, the PMR subset/subgroup/cluster is to be treated in accordance with existing PMR recommendations, although it may have atypical manifestations and a shorter course.

Nevertheless, only a few studies documented features that would warrant the diagnosis of a subset of the disease, at least in accordance with our entry definitions. This is certainly true for PMR with normal baseline APRs, as all the analyzed studies agree on excluding an alternative diagnosis [6,7,31].

With regards to PMR following infections, and PMR following vaccines, available data point in the direction of PMR subsets [23,47,48,51], except for very few studies in which ASIA syndrome is suspected [38,39]. However, the majority of studies listed in Table 2 and Table 3 are within small case series and—consequently—should be investigated in further ad hoc studies.

The relationship between immunotherapy with ICIs and PMR is still lacking a definite framework [100]. To date, many researchers have argued that ICI-PMR is a PMR-mimicking condition. However, the possibility that ICI-PMR can be a subset of disease cannot be categorically excluded. As a matter of fact, to date, ICI-PMR has not been described in ICI patients aged under 50 years, an age below which the diagnosis of primary PMR is very questionable, according to all the published and validated diagnostic and/or classification criteria. In addition, several cohort studies of ICI-treated patients agree that the incidence of ICI-PMR is up to 1%, an incidence rate much higher than the estimated incidence rate (0.1%/year) for primary PMR in age-appropriate populations. This is likely to be an underestimation if we accept the likelihood of underdiagnosis of PMR by oncologists. Finally, it is true that some ICI-PMR patients may have a self-limiting and monophasic course persisting for a far shorter duration than primary PMR. However, it is also possible that patients with ICI-PMR may have a chronic course similar to that of primary PMR [101].

To date, primary PMR is considered a macrophage-focused pathology. On the other hand, ICIs, by blocking checkpoint molecules such as CTLA -4, PD -1, and PD -L1, disturb the equilibrium of immune tolerance, possibly triggering an unregulated T-cell attack on self-antigens [108,109]. Given this background, primary PMR and ICI-PMR should be considered different diseases. However, the pathogenic path does not necessarily have to be dichotomous. Indeed, it could be hypothesized that in ICI-PMR, the first trigger is represented by an antigenic stimulus (potentially activated by the primary or metastatic tumor mass) recognized by the antigen-presenting macrophages. Subsequent activation of T-lymphocytes induced by ICIs could favor their infiltration in the anatomical sites where PMR starts [109].

Without a doubt, the lack of a validated definition of ICI-PMR can cause confusion in the categorization of this entity. Moreover, another methodological grey area could be the modality for assessing ICI-PMR as an adverse drug reaction (ADR). Specifically, applications of validated scales such as Naranjo’s scale for the identification of ICI-PMR as an ADR are still lacking in the published literature. Clinical judgement is still an unappealable criterion, with all the limitations that this may entail [110]. As recently confirmed by a EULAR/ACR task force, using the Naranjo scale may help to assess the causal link between rheumatologic immune-related adverse events (IRAEs) and ICI therapy [111].

Notably, no cases of GCA were diagnosed during the follow-ups of patients with ICI-PMR. Some authors reported on the low expression of some checkpoint inhibitors (specifically, PD-1 and PD-L1) within the temporal arteries of patients with GCA [112]. This apparent contradiction is still awaiting a convincing explanation.

Similarly, the relationship between PMR and CPPD disease is still being discussed. The study by Pego-Reigosa in 2005 proposed to include CPPD among the rheumatic diseases with which PMR can be confused. The authors suggested the so-called “pseudo-PMR pattern” of CPPD, defining a different diagnosis from that of pure PMR [76]. Most studies suggest a more favorable course for this subset of patients [17,18,19,76,80]. However, as the prevalence of CPPD increases with age, its random coexistence with PMR is possible, and some authors suggest to not definitively exclude PMR diagnosis despite the presence of chondrocalcinosis on imaging [74,82,113]. In clinical practice, a change in diagnosis from PMR to CPPD should probably depend on discerning if CPPD crystals can be considered responsible for the pathology and related symptoms. In this context, the recent EULAR recommendations on CPPD terminology and diagnosis [114] stated that a definitive diagnosis of CPPD relies on the identification of CPP crystals on SF (positively birefringent intra- or extracellular calcium crystals), whereas inflammatory symptoms and signs with concurrent CC are suggestive but not definitive of acute or chronic CPP crystal arthritis. On the other hand, Filippou and Sirotti suggest that the mere presence of “asymptomatic CPPD” should be regarded as a “preclinical stage of CPPD” or probably “early CPPD” without joint inflammation that permits an SFA, but with a typical US appearance on specific joints (at least triangular fibrocartilage of wrists and knees) [115].

Radiographic CC is not highly sensitive or specific, whereas ultrasonography (US) appears more useful for diagnosis [115,116]. Moreover, the recent ACR/EULAR CPPD classification criteria state that both the presence of either crowned dens syndrome or synovial fluid analysis demonstrating CPP crystals in a symptomatic joint is a sufficient criterion for CPPD classification [117]. On the other hand, in other cases with a polyarticular and rhizomelic PMR-like presentation, CPPD diagnosis and classification requires in-depth study of clinical history and imaging [118]. Considering the absence of natural history studies on CPPD, we cannot exclude that “asymptomatic chondrocalcinosis” or “preclinical CPPD” could have its first clinical presentation as a PMR syndrome with prevalent rhizomelic symptoms. At the same time, we have not determined the definite prevalence of axial involvement in asymptomatic CPPD, but it is possible that a milder form of CDS could be responsible for neck and shoulder stiffness in this peculiar subset.

Independently from the debate on if the co-existence of CPPD could be pathogenetic in polymyalgia syndrome, an interesting aspect should be underlined regarding the prognosis and clinical course of patients with PMR and CPPD. In fact, in all the studies on this topic, and in the long-term study by Conticini et al. [19], no patient with PMR/CPPD had a late diagnosis of GCA (at least within the follow-up window).

This remark leads to the final point of discussion: GCA was never diagnosed during the follow-ups of the patients falling into the PMR subsets/subgroups/clusters identified in our review. To date, a discussion on the role that triggering factors may have in the so-called “spectrum PMR/GCA disease” is highly speculative [119]. Should the working concept of the spectrum of PMR/GCA disease be applied only to classical PMR and not to subsets of PMR? What emerged from our literature search seems to be moving in this direction. In addition, when a patient first diagnosed or categorized as PMR also shows evidence of having subGCA, this patient should be managed according to the most severe condition. Consequently, a final diagnosis of GCA should be made and the case categorized as a PMR-mimicking condition.

Our literature search has limitations and strengths. To the best of our knowledge, this is the first review that takes into account all the data present in the published literature in a non-sectoral vision, in order to identify possible PMR subsets/subgroups/clusters and to differentiate them from PMR-mimicking conditions. The heterogeneity of the available data, use of local and not validated protocols, and lack of data on some topics were the significant limitations we found in the published literature.

## 5. Conclusions

Our narrative review provided an in-depth overview of everything present in the published literature about subsets of PMR and the most common PMR-mimicking conditions.

Recognizing PMR subsets or PMR-mimicking conditions does not just carry nomenclature value and speculative significance. Indeed, it can also have repercussions on epidemiological assessments, health policies, prognosis and therapeutic approach.

According to the entry definitions we proposed, PMR with normal baseline ESR and CRP concentrations, post-infection PMR and PMR following vaccination (with the exception of rare and questionable ASIA syndromes) should be categorized as subsets of disease. PMR/subclinical GCA and most cases of PMR/CPPD should be categorized as mimickers, but adequate and early imaging seems to be mandatory to define these conditions.

Lastly, the question of whether or not ICI-PMR should be categorized as a mimicker or as a subset of the disease is still awaiting a definitive answer., Moreover, further studies are required to better categorize the two clinical subsets emerging from cluster analyses (that is, younger patients/high inflammation and older patients/low inflammation).

Immune–histopathological studies are definitely needed to further enhance our current knowledge on this topic, favoring the more accurate categorization of all patients with PMR.

Finally, PMR was sometimes diagnosed using local protocols, without referring to internationally validated criteria. We hope that this methodological approach will be dismissed in the future, thus allowing for greater comparability among the data.

## Figures and Tables

**Table 1 healthcare-13-01226-t001:** Polymyalgia rheumatica with normal baseline acute-phase reactants.

Reference	Study Design	Study Sample (Peculiar PMR Patients/Total Sample)	Diagnosis of PMR	Length of Follow-Up	Imaging	Presence of Definition of Subset/Subgroup/Cluster	Significant Characteristics of Subset/Subgroup/Cluster	Suggested PMR-like Condition/Other Nosology Entity
Manzo et al. [7]	Retrospective study	7/460 (1.52%)	Healey and ACR/EULAR criteria	29–120 months	US and 18-FDG PET/CT	Yes, subset of PMR patients with normal baseline acute-phase reactants	Absence of systemic manifestations (systemic manifestations in one patient)	No alternative diagnosis
Marsman et al. [6]	Retrospective cohort study	62/454 (13.6%)	Local protocol	10-year (on average)	US and MRI	Yes, subset of PMR patients with normal baseline acute-phase reactants	PMR with a milder presentation: longer median symptom duration before diagnosis, younger age, lower comorbidities and systemic symptoms	None; no GCA was reported
Kara et al. [31]	Retrospective study	8/54 (14.8%)	2012 EULAR/ACR criteria (declared); local protocol (in the essence of the facts)	At least 1 year	US, MRI and PET/CT for selected patients	Yes, subset of PMR patients with normal baseline acute-phase reactants	longer median duration of symptoms, lower anemia, no differences in systemic symptoms, no differences in response to GCs after 4 weeks	None; no alternative diagnosis

PMR = polymyalgia rheumatica, ACR/EULAR = European League Against Rheumatism/American College of Rheumatology, MRI = magnetic resonance imaging, ^18^F-FDG PET-CT = ^18^F-labeled fluorodeoxyglucose-positron emission tomography-computed tomography, US = ultrasound, and GC = glucocorticoids.

**Table 2 healthcare-13-01226-t002:** Polymyalgia rheumatica with infection trigger.

Reference	Study Design	Study Sample (Peculiar PMR Patients/Total Sample)	Diagnosis of PMR	Length of Follow-Up	Imaging	Presence of Definition of Subset/Subgroup/Cluster	Significant Characteristics of Subset/Subgroup/Cluster	Suggested PMR-like Condition/Other Nosologic Entity
Falsetti et al. [23]	Retrospective Study	9/58 (15.5%)	Bird and ACR/EULAR criteria	At least 24 months	US	Yes, PMR patients reporting a previous infection before the onset of the disease	Higher CRP, faster response to GCs, milder shoulder synovitis	No
Duarte Salazar et al. [35]	Case Report	1	ACR/EULAR criteria	6 months	US	Yes, PMR patient reporting a previous infection before the onset of the disease	Longer median symptom duration before diagnosis, younger age, lower comorbidities and systemic symptoms, normal CRP values, positive PCR test result for COVID-19, faster response to GCs, complete recovery at 6 months of follow-up	No
Ursini et al. [36]	Observational study	28/122 (22.9%)	ACR/EULAR criteria	Not clear	Not reported	Yes, PMR patients reporting a previous infection before the onset of the disease	longer median duration of symptoms, lower anemia, no differences in systemic symptoms, no differences in response to GCs after 4 weeks	No

PMR = polymyalgia rheumatica, ACR/EULAR = European League Against Rheumatism/American College of Rheumatology, US = ultrasound, GC = glucocorticoids, CRP = C-reactive protein, and COVID-19 = coronavirus disease 2019.

**Table 3 healthcare-13-01226-t003:** Polymyalgia rheumatica following vaccination.

Reference	Study Design	Study Sample (Peculiar PMR Patients/Total Sample)	Diagnosis of PMR	Length of Follow-Up	Imaging	Presence of Definition of Subset/Subgroup/Cluster	Significant Characteristics of Subset/Subgroup/Cluster	Suggested PMR-like Condition/Other Nosological Entity
Soriano et al. [38]	case series	10 GCA/PMR post inf-V (2 pure PMR)	Healey criteria	na	na	GCA/PMR within 3 months by inf-V		possible (suggestion for ASIA)
Bassendine et al. [39]	case report	1 PMR relapse after ADJ-infV	na	8 months	na	flares after ADJ-infV	girdle pain and knee arthritis, atypical	yes, ASIA
Falsetti et al. [23]	retrospective mono-centric	58 cases PMR following environmental trigger	Bird and ACR/EULAR criteria	2 years	US	yes: PMR patients describing an environmental trigger that occurred within 3 months from the onset of PMR, judging it as correlated to the symptoms	subset “PMR with environmental trigger”: higher CRP at onset, lower frequency of gleno-humeral synovitis on US, and shorter time to normalize inflammatory reactants, but higher frequency of GC dependence	No
Liozon et al. [40]	case series and review	12 pz case series 358 reviewed	ACR/EULAR criteria		na	yes: GCA or PMR within 1 month from influenza vaccination	PMR post-InfV: self-limited GCA post-InfV: more protracted course (chronic relapsing disease in one-third of patients	no
Manzo et al. [41]	case report	1 PMR after mRNA vaccine	ACR/EULAR criteria	5 months	^18^F-FDG PET-CT and US	PMR after COVID-19 mRNA vaccine	typical	no
Liozon et al. [42]	case series	5 PMR	ACR/EULAR criteria	until 9 months	na	yes: PMR within 3 weeks from COVID-19 vaccination	clinical presentation and prognosis not different with respect to PMR without triggers; however, not self-limited or benign	not specified
Ottaviani et al. [43]	case series	10 new-onset PMR after COVID-19 vaccination	ACR/EULAR criteria	10.5 weeks (range 3–24)	^18^F-FDG PET-CT and US	yes: PMR within 2 weeks (range: 5–15 days) from COVID-19 vaccination	clinical presentation and prognosis not different with respect to PMR without triggers	no
Mettler et al. [44]	VigiBase database pharmaco-vigilance SAR (suspected adverse reaction) post-COVID-19 or Inf-V vaccines	290 PMR over 1,295,482 COVID-V (0.022%) 303 PMR over 317,687 Inf-V	na	na	na	PMR within 14 days from vaccines	no differences in age and gender: seriousness, 57.2%; recovered, 8.3%; not recovered, 28.6%	no
Rider et al. [45]	GRA vaccine survey	197 PMR-vaccinated; flares in 16 (5.8%); prevalence 8.1%	na	na	na	flares % in PMR vaccinated	OR 2.71 for females, more frequent for Astra-Zeneca, and previous reactions; no correlation with age, smoking, or therapy; no definite subset	no
Carubbi et al. [46]	case series	153 vaccinated; 108 no previous disease; 4 PMR (11%) +1 flare (2%)	na	na	na	new PMR after 3 doses mRNA vaccine	No definite subset	no
Ursini et al. [36]	case series	46 PMR post-COVID-19 vaccinations	na	14 weeks (±13)	na	yes: PMR within 1 month from COVID-19 vaccination	PMR post-vaccines: 74% well responders to first-line therapy; mean age slightly inferior to typical PMR (62 years), with 6/46 aged < 50	no, possible misclassification for age < 50
Bandinelli et al. [47]	retrospective mono-centric	177 rheumatic symptoms post-COVID-19 vaccines; 109 included, 22 PMR and GCA	ACR/EULAR criteria	6 months	US PDUS	yes: PMR within 1 month from COVID-19 vaccination	frequently females (81.8%); age, 71 y; remission: 3 months, 45.4%; 6 months 90.9% “notably elevated percentage of remission observed after six months”, “lower count of natural killer cells, CRP and ESR higher than undifferentiated arthritis	no
Haruna et al. [48]	case report	1 PMR case post-COVID-19 vaccine	Chuang and Healey criteria	na	US, ^18^F-FDG PET-CT	yes: PMR within 1 month from COVID-19 vaccination	high ESR and CRP, rapid response to low-dosage GC no involvement of shoulders on US and PET-CT, only pelvic girdle involvement	no
Furr et al. [49]	case report	2 PMR cases after COVID-19 vaccines	na	8 m	na	new PMR after COVID-19 mRNA vaccines	typical	no
Pinto Oliveira et al. [50]	database pharmaco-vigilance SAR post-COVID-19 vaccines	433 suspected PMR cases out of 1,426,786 (0.03%)	na	na	na	ICSR signaled by healthcare professionals within the European Economic Area, containing a SAR of PMR between 1 January 2021, and 1 May 2023, attributed to COVID-19 vaccines approved by the EMA	non-recovered 44.8%	no
Jarrot et al. [51]	case series	60 PMR post-COVID-19 vaccinations	ACR/EULAR criteria	16 m (range 12–20)	US and ^18^F-FDG PET-CT	yes: PMR within 1 month from COVID-19 vaccination	Tapering GC schedule shorter than recommended (mean duration 8 months), slightly lower relapse rate of 10% in first year, vs. 20–55% for unvaccinated cases	no
Kim et al. [52]	WHO database	25219 AEFIs 2398 post-COVID-19 vaccine; 581 after other vaccines	na	na	na	PMR onset within 4 days (range: 1–11) after COVID-19 vaccines	PMR (ROR 1.42); a significant number of patients with AEFI did not recover (5261, 20.9%) or had sequelae (655, 2.6%) (global data); no definite PMR subset.	no

PMR = polymyalgia rheumatica, GCA = giant cell arteritis, ACR/EULAR = European League Against Rheumatism/American College of Rheumatology, US = ultrasound, PDUS = power doppler ultrasound, GC = glucocorticoids, ESR = erythro-sedimentation rate, CRP = C-reactive protein, COVID-19 = coronavirus disease 2019. MRI = magnetic resonance imaging, ^18^F-FDG PET-CT = ^18^F-labeled fluorodeoxyglucose–positron emission tomography–computed tomography, US = ultrasound, GCs = glucocorticoids, na = not assessed, ASIA = autoimmune/inflammatory syndrome induced by adjuvants, ADJ-inf-V = adjuvated influenza vaccine, SAR = suspected adverse reaction, AEFI = adverse event following immunization, ICSR = individual case safety reports, WHO = World Health Organization, and EMA = European Medicines Agency.

**Table 5 healthcare-13-01226-t005:** Polymyalgia rheumatica and calcium pyrophosphate deposition disease.

Reference	Study Design	Study Sample (Peculiar PMR Patients/Total Sample)	Diagnosis of PMR	Length of Follow-Up	Diagnostic Tools	Presence of Definition of Subset/Subgroup/Cluster	Significant Characteristics of Subset/Subgroup/Cluster	Suggested PMR-like Condition/Other Nosologic Entity
Dieppe et al. [74]	prospective monocentric	8 PMR/CPPD over 105 CPPD	na	5 years of GC therapy	clinical and CR	CPPD with concurrent PMR	CPPD/PMR: advanced age	coexistence of PMR and CPPD for chance association or long-term GC therapy
Aouba et al. [75]	case series	5 CDS PMR-like	clinical	Max. 14 months	clinical, CR and CT	CDS-CPPD with PMR-like presentation	older, CC in Rx and CT, CDS in atlo-axial CT. Responsive to NSAIDs and/or colchicine	yes, different diagnosis and therapy between CPPD and PMR
Pego-Reigosa et al. [76]	prospective monocentric	118 PMR; 36/118 (31%) CPPD/PMR	PMR: Chuang and Healey criteria CPPD: Mc Carty criteria	1 year	radiologic imaging and SFA	CPPD with concurrent PMR	CPPD/PMR: 31%, older, more frequent peripheral arthritis, more advanced knee osteoarthritis, more frequent tendinous calcifications and ankle and wrist arthritis. Shorter GC course and disease duration (not significant.	yes, shorter GC course and disease duration (not significant)
Salaffi et al. [77]	case series	2 PMR-like over 25 CPPD with CDS	CPPD: Mc Carty criteria		clinical and CT	CDS-CPPD with PMR-like presentation	older, chondrocalcinosis in Rx and CT, CDS in atlo-axial CT	yes, different diagnosis between CPPD and PMR
Yanai et al. [78]	case series	1 CPPD over 10 PMR	SFA		SFA and CR	CPPD with PMR-like presentation	older, higher CRP, prompt response to NSAIDs	yes, different diagnosis and therapy between CPPD and PMR
Siau et al. [79]	case report	1 PMR-like CPPD	clinical	na	CT	CDS-CPPD with PMR-like presentation	higher CRP, prompt response to NSAIDs and GC	yes, different diagnosis between CPPD and PMR
Ceccato et al. [80]	retrospective multicentric	200 PMR syndrome; 16/200 (8%) other diagnosis in follow-up; 2 CPPD (CDS)	Chuang criteria	1 year	CR	CPPD with concurrent PMR	PMR/CPPD: 1%, calcifications C1C2, typical chondrocalcinosis, good response to NSAID	yes, other diagnosis with different therapy and better outcome
Falsetti et al. [18]	prospective monocentric	61 PMR at onset; 9/61 (15%) PMR/CPPD	Bird criteria	1 year	US	PMR with US diagnosis of other conditions	PMR/CPPD: 15%, more frequent in females, higher frequency of knee menisci calcifications and tendinous calcaneal calcifications, and lower ESR, CRP, and PLT vs. others	yes, suggestion of a lower dosage of GC and different course
Oka et al. [81]	case series and review	72 CDS published cases, 7 (19.4%) PMR-like			CT	CDS-CPPD wit PMR-like presentation	higher CRP (mean 12.6 mg/dL), frequent fever (80.4%), prompt response to only NSAIDs 67.5%	yes, different diagnosis and therapy between CPPD and PMR
Manzo et al. [82]	prospective monocentric	134 PMR syndrome (by general practictioners) 41 PMR 93 not-PMR	Healey	18 months	not specified	PMR syndrome with following diagnosis of other conditions	PMR/CPPD: 11%	yes, but Authors consider the possible overlap between the two diseases
Ottaviani et al. [17]	prospective monocentric	94 PMR syndrome; 52 PMR diagnosis; 25/52 (48%) PMR/CPPD	ACR/EULAR criteria and McCarty/Zhang ACR/EULAR recommendations	na	US and SFA	yes: PMR patients with concurrent diagnosis of CPPD (on US and synovial fluid analysis)	PMR/CPPD: 48%, more frequent in older females, with humeral head erosions, synovitis and calcifications of AC joint; lower frequency of SAD bursitis	yes, PMR/CPPD is considered another condition, suggesting shorter courses of GC
Conticini et al. [19]	retrospective multicentric	204 PMR syndrome; 31 CPPD out of 104 evaluated by US (22%)	Bird and ACR/EULAR criteria	12–60 months	US e CDUS vs. only clinical	yes: PMR syndrome with early or late diagnostic shift to other diagnosis	PMR/CPPD: 22%, patients with more frequent peripheral synovitis, lesser frequent flares, lesser dependence on GC, more frequent use of DMARDs.	yes, PMR/CPPD is considered another condition, suggesting different management; only PMR with US at onset can undergo a change upon diagnosis
Ono et al. [83]	case series	4 PMR patients with subsequent diagnosis of CPPD	not clear	6–24 months	Clinical, CT, SFA	yes: PMR patients with subsequent diagnosis of CPPD	PMR/CPPD: more frequent peripheral arthritis, usefulness of NSAID and Colchicine in management	yes, PMR/CPPD is considered another condition, suggesting a different management approach

PMR = polymyalgia rheumatica, CPPD = calcium pyrophosphate deposition disease, CDS = crowned dens syndrome, ACR/EULAR = European League Against Rheumatism/American College of Rheumatology, US = ultrasound, CDUS = color doppler ultrasound, CR = conventional radiography, GC = glucocorticoids, ESR = erythro-sedimentation rate, CRP = C-reactive protein, SFA = synovial fluid analysis, MRI = magnetic resonance imaging, CT = computed tomography, na = not assessed, NSAID = non-steroid anti-inflammatory drugs.

**Table 6 healthcare-13-01226-t006:** Polymyalgia rheumatica following immunotherapy with immune checkpoint inhibitor drugs.

Reference	Study Design	Study SAMPLE (Peculiar PMR Patients/Total Sample)	Diagnosis of PMR	Length of Follow-Up	Imaging	Presence of Definition of Subset/Subgroup/Cluster	Significant Characteristics of Subset/Subgroup/Cluster	Suggested PMR-like Condition/Other Nosological Entity
Belkhir et al. [85]	Retrospective	4	ACR/EULAR criteria	Not reported	None	No.	No suggestion.	
Kuswanto et al. [86]	Retrospective	4	Not clear	Not reported		None. PMR-like conditions and/or peripheral synovitis.		
Kostine et al. [87]	Prospective observational	11	Not clear	Not reported			Patients with rheumatic irAEs had a higher tumor response rate compared with patients without irAEs (85.7% vs. 35.3%; *p* < 0.000.	Association with GCA not reported/specified.
Leipe et al. [88]	Prospective cohort study	5 pz. with PMR and concurrent arthritis (oligoarthritis: 1 pz; poliarthritis: 2 pz; monoarthritis: 2 pz). Monarthritis presented as omarthritis in all cases	Not reported	The entire patient cohort was followed-up on for a median of 433 days	US, MRI, PET-CT scans	None.	Not specified.	Most of the data of interest to us are missing in this article. Consequently, a specific assessment was not possible.
Salem et al. [89]	Observational study based on Vigibase	16 PMR pz. among 31,321 overall immunotherapy: 0,05% 14/16 following mono-PD1 (11–nivolumab)	Not specified	Not specified	Not reported			VigiBase is the WHO’s global Individual-Case-Safety-Report (ICSR) database to identify drug-AE related to ICIs. Messages from databases can be misleading, and should be critically assessed.
Calabrese et al. [90]	Three centers Retrospective	37 (only 12 fulfilling classification criteria)	ACR/EULAR criteria	Not clear	Not reported.	Their overall clinical picture suggested that at least some may represent a new clinical entity.	Atypical features.More severe presentations than generally encountered in classical PMR; 37% of cases required more aggressive therapy with GC than is traditionally used to treat PMR. Seven patients had normal acute-phase reactants at the time of PMR diagnosis.	Their overall clinical picture suggested that at least some manifestations may represent a new clinical entity.
Richter et al. [91]	Monocentric retrospective cohort study	4/61. 1 case of PMR flare and 3 cases of new-onset PMR-like syndromes.	Not reported	Not reported	Not reported	No.	Resolution in 1 pz; chronic symptoms in 3 patients.	
Roberts et al. [92]	Retrospective cohort study	17 (anti-PD1: 9)	Clinical judgment	3–50 months	Not reported	None.		
Allenbach et al. [93]	Observational study based on Vigibase	76 pz with PMR/54,416 overall ICIs (69 following mono-PD1)	Not specified	Not specified	Not reported			Allenbach et al.’ s data suggest that ICI-PMR may have different pathophysiological mechanisms as compared to idiopathic PMR
de Fremont et al. [94]	Case–control study: 14 ICI-PMR vs. 43 classical PMR cases	14 pz. with inflammatory arthritis mimicking PMR (11 fulfilling classification criteria)	ACR/EULAR criteria	Not specified	5 patients in the ICI group underwent ^18^F-FDG PET/CT imaging before rheumatologic treatment	No definition.	Higher prevalence of peripheral arthritis in ICI-PMR (57.1) vs. 43 pz with “classical” PMR (27.9%); difference in sex ratio (14.3% women in ICI-PMR group vs. 39.5% women in classical PMR).	The therapeutic strategies remain the same as what is proposed in classical PMR. Association with GCA not specified.
Gomez-Puerta et al. [95]	Observational study	10 patients	Clinical judgment	Mean time follow-up was 14.0 ± 10.8 (months)	Not reported	None.	Not specified.	PMR treatment did not appear to negatively impact the tumor response to immunotherapy.
Ponce et al. [96]	retrospective observational study	5 pz (in one patient, time from nivolumab and PMR-like manifestations = 6 months)	Not specified	Not clear	US, MRI, PET/CT scans	Not proposed.	ICI-induced PMR has US and FDG-PET/CT results comparable to those seen in idiopathic PMR. In treated patients, no synovitis, tenosynovitis, or bursitis in the shoulders and pelvic girdles were found. They seemed to have resolved with GC treatment. Interestingly, peripheral synovitis/tenosynovitis in the hand joints remained evident. Many patients have mild clinical symptoms.	
Kato et al. [97]	observational study based on the data recorded in the Japanese Adverse Drug Event Report (JADER) database.	67	Not specified	Not specified	Not reported			PMR was not clearly distinguished from GCA (the authors wrote that “*PMR is characterized by GCA and inflammatory symptoms*”) and references were often misleading.
Ceccarelli et al. [98]	descriptive study	6 pz. (N.B.: in one patient, PMR was diagnosed with a 112-week [about 2 years] interval following therapy with nivolumab. Casual association?)	ACR/EULAR criteria	Not clear	US	None.		
Vermeulen et al. [99]	monocentric retrospective case-control study:	15 ICI-PMR in patients with cancer vs. 37 idiopathic, primary PMR	Chuang and ACR/EULAR criteria	a maximum of 990 days after start of glucocorticoid treatment	US, ^18^F-FDG-PET/CT	ICI-PMR is associated with less intense inflammation than primary PMR. This was further substantiated by the milder disease course and lower treatment requirement (usually <10 mg/day) observed in the ICI-PMR patients.	ICI-PMR is associated with less intense inflammation than primary PMR. This was further substantiated by the milder disease course and lower treatment requirement (usually <10 mg/day) observed in the ICI-PMR patients.	Although ICI-PMR and primary PMR share a cluster of symptoms related to inflammation in the shoulder and hip girdle, ICI-PMR is a different disease entity than primary PMR. GCA is not associated with ICI-PMR (0%).
Hysa et al. [25]	systematic review	314	The ACR/EULAR criteria for PMR were utilized in 1 case, 2/4 observational studies (50%) and 9/14 case reports (64%). The remaining studies relied only on physicians’ clinical judgment for diagnosis.	2 years	US, CT, MRI	ICI-PMR as a PMR-like syndrome.	Laboratory tests showed normal or slightly elevated inflammatory markers in 26% of cases. GCs led to symptom improvement in 84% of cases, although 20% required immunosuppressive treatment and 14% experienced relapses.	

PMR = polymyalgia rheumatica, ICI = immune checkpoint inhibitor, GCA = Giant cell arteritis, ACR/EULAR = European League Against Rheumatism/American College of Rheumatology, US = ultrasound, PDUS = power doppler ultrasound, GC = glucocorticoids, ESR = erythro-sedimentation rate, CRP = C-reactive protein, COVID-19 = coronavirus disease 2019. MRI = magnetic resonance imaging, ^18^F-FDG PET-CT = ^18^F-labeled fluorodeoxyglucose–positron emission tomography–computed tomography, US = ultrasound, GC = glucocorticoids, na = not assessed.

**Table 7 healthcare-13-01226-t007:** Polymyalgia rheumatica with peculiar clinical subsets.

Reference	Study Design	Study Sample (Peculiar PMR Patients)	Diagnosis of PMR	Length of Follow-Up	Imaging	Presence of Definition of Subset/Subgroup/Cluster	Significant Characteristics of Subset/Subgroup/Cluster	Suggested PMR-like Condition/Other Nosologic Entity
Gonzalez-Gay et al. [102]	case series	4 PMR	Hunder and Haley criteria	1 year	no, only clinical	yes: PMR patients with other diagnoses made late	PMR-like conditions: unsatisfactory response to low-dosage GC, small joint synovitis, fever, monoarthritis, lymphadenopathy	yes: PMR-like conditions considered to be other nosologic conditions
Mackie et al. [103]	prospective monocentric	176 PMR (124 stopped GC)	Bird criteria	5 years	na	Yes; PMR able to stop GC after 5 years	subset requiring GC and late GCA: female, higher GC > 15 mg onset, PV (ESR) > 2 onset, history weight loss	no; suggestion for increased adrenal suppression by higher dosage of GC, in patients with previous adrenal impairment
Mackie et al. [104]	prospective monocentric	22 PMR	Bird crieria	2 years (median)	whole-body MRI	yes: (“extracapsular pattern” described by whole-body-MRI	extracapsular pattern: males, higher CRP, higher IL-6, better HAQ-DI and fatigue-VAS at onset, better and faster response to GC, but requiring GC treatment for >1 year.	No
Quartuccio et al. [105]	retrospective monocentric	100 PMR with MTX	clinically judged by a rheumatologist and a posteriori ACR/EULAR criteria	12–185 months (median 46.5 months)	na	Yes: clinical subgroups derived from MTX introduction criteria. Group A: patients with relapse of PMR during the 1st month of therapy, when tapering off of glucocorticoids (GCs) Group B: patients requiring long-term GC therapy; Group C: patients still requiring >5 mg/day of GC after 4 months; Group D: patients with GC-related side effects; Group E: patients at high risk of GC-related side-effects.	no significant differences were noticed among the 5 subgroups, with regard to all the outcomes measured. Compared with the GC-alone group, the MTX group had patients of a younger age, and had a higher prevalence of female patients with a higher level of inflammation.	no
Hayashi et al. [106]	prospective monocentric	61 PMR	ACR/EULAR criteria	until 2 years (21 months ±6)	US in only 10 patients	yes: hierarchical cluster analysis (Ward’s method) from 5 selected variables at onset	Cluster 1 “with thrombocytosis”: higher PLT, rare peripheral arthritis, worse response to treatment, more frequent refractory cases, requiring DMARDs. Cluster 2 “without peripheral manifestations”: lower WBC, lower morning stiffness, no peripheral synovitis, with persistent PMR diagnosis and only GC therapy. Cluster 3 “with peripheral arthritis”: more points in RA criteria score.	No
Falsetti et al. [23]	retrospective monocentric	58 PMR	Bird criteria and ACR/EULAR criteria	2 years	US	yes: PMR patients describing an environmental trigger that occurred within 3 months from the onset of PMR and judging it as correlated to the symptoms	subset “PMR with environmental trigger”: higher CRP at onset, lower frequency of gleno-humeral synovitis on US, shorter time to normalize inflammatory reactants, but higher frequency of GC dependence	No
Muller et al. [9]	prospective multicentric	652 PMR	clinically judged by a general practitioner	2 years	na	yes: clinical clusters derived from latent class growth analysis (LCGA), which is a data-driven approach used to estimate the trajectory of pain and stiffness. Cluster 1: sustained symptoms. Cluster 2: partial recovery with sustained moderate symptoms. Cluster 3: recovery before worsening. Cluster 4: rapid and sustained recovery. Cluster 5: slow and continuous recovery.	Cluster 1: poorer health at baseline, higher dose of GCs after 1 year, more frequent referrals to a specialist. Cluster 3: possible other condition. Cluster 4: better health at baseline, greater frequency of males, longer-lasting morning stiffness, higher fatigue at baseline, more persistent PMR diagnosis	yes, possible mimickers and change in diagnosis for Cluster 3
Falsetti et al. [107]	case report	2 PMR	ACR/EULAR criteria	na	US/PDUS	yes: co-existence of capsulitic/enthesitic features on shoulder PDUS in PMR patients	Early-PMR patients: capsulitic/enthesitic process of the ligamentous and capsular structures (coraco-humeral pulley and superior gleno-humeral ligament) within rotator interval	No
Colaci et al. [71]	retrospective monocentric	17 PMR with persistent inflammation, over 80 PMR	ACR/EULAR criteria	At least 1 year	^18^F-FDG PET/CT	yes: PMR patients who underwent ^18^F-FDG-PET/CT because of the persistent increase in acute-phase reactants (APRs) besides steroid therapy	More frequently female patients, higher CRP and ESR, higher grades of articular and periarticular inflammation (suggesting chronic arthritis), subclinical GCA in about a third of cases	Yes, PMR with persistent increases in APR are probably chronic arthritis or subclinical GCA
Conticini et al. [19]	retrospective multicentric	201 PMR	Bird criteria and ACR/EULAR criteria	until 5 years	US/PDUS and CDUS	Yes: cluster analysis from all collected continuous variables at onset	Cluster 2: older patients, lower systemic inflammation, lower WBCs, PLT and Hb, higher persistence of PMR diagnosis (no diagnostic shift), more frequent shoulder pain and tenderness, lower PD signals in shoulders and wrists, less peripheral synovitis, more common environmental triggers before onset (vaccination), more frequent flares and greater likelihood of requiring GCs at 1 and 2 years.	No

PMR = polymyalgia rheumatica, GCA = giant cell arteritis, ACR/EULAR = European League Against Rheumatism/American College of Rheumatology, US = ultrasound, PDUS = power doppler ultrasound, CDUS = color doppler ultrasound, GC = glucocorticoids, ESR = erythro-sedimentation rate, CRP = C-reactive protein, PV = plasma viscosity, COVID-19 = coronavirus disease 2019. MRI = magnetic resonance imaging, ^18^F-FDG PET-CT = ^18^F-labeled fluorodeoxyglucose–positron emission tomography–computed tomography, TAB = temporal artery biopsy, US = ultrasound, GC = glucocorticoids, na = not assessed, LVV = large vessel vasculitis, LBP = low back pain.

## Data Availability

Not applicable.

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
