# Peer review of "Are There Definite Disease Subsets in Polymyalgia Rheumatica? Suggestions from a Narrative Review"

_healthcare, 2025, doi:10.3390/healthcare13111226_

Round 1

Reviewer 1 Report

Comments and Suggestions for Authors

Dear authors,

Your manuscript addresses a subject of great interest. It is timely and relevant. However, I must say I missed a more systematic approach to the extraction and analysis of data. In the end, I felt I was dealing with a lot of heterogeneous information, without means of comparing or judging it. Because of the enormous amount of information being delivered, consider making it more didactic or synthetic for the readers.

Please find my issues and suggestions below.

Best of luck.

- In “Half patients with initial PMR manifestations may experience a change in diagnosis at follow-up, often towards chronic arthritis,” consider checking the grammar.

- The part of “Definitions” in the Introduction is somewhat displaced in my opinion. In fact, I am not sure you actually applied your definitions throughout the study… If you believe these definitions are worthy to be mentioned, consider addressing them on the Discussion section.

- Considering your definitions, you have chosen to apply the Bird’s criteria because of sensitivity and the ACR/EULAR criteria because of specificity. But, instead of adding those, you decided to choose one of them. Considering your goal—to isolate the cases with the greatest certainty of true PMR—should we not use the combination of both—adding, thus, sensitivity and specificity?

- On the other hand, when searching for PMR-mimics, should we not use the most specific criteria (i.e.: reducing the false-positive)?

- You stated that “Searches were performed regardless of language” but the exclusion criteria appear to have excluded non-English articles. So, you searched non-English articles only to exclude them later?

- Isn’t it curious that patients with normal APR have less systemic symptoms? Consider discussing this apparent paradox.

- How did Kara et al. included patients with PMR without elevated APR using the ACR criteria if elevated APR are required for entering the criteria?

- In order to facilitate understanding for the readers and to allow them to judge the scientific soundness, could you provide the criteria Marsman et al. utilized in their study? Considering they are not validated, should they even be used?

- This is just a detail, but temporal artery biopsy is not actually an imaging modality. Consider adjusting the column heading.

- While the tables provide a lot of content, I found it very tiresome to navigate all of them. Maybe you could consider another tool for showcasing those results (a graph or illustration, for instance). You can maintain the tables as supplementary for readers interested in more details.

- In the studies within the ICI-PMR section, many reports rely on clinical judgement or no criteria whatsoever for diagnosing PMR. It is reasonable to use clinical judgement in daily clinical practice, but regarding scientific reporting, I understand that those articles defy the definition of “scientific”. Should we really be reporting them? Consider addressing this issue in your discussion.

- Consider checking the syntactic parallelism in: “a second with lesser systemic and articular inflammation, with persistence in PMR diagnosis and only GC.”

- Please cite the references supporting: “To date, primary PMR is considered a macrophage-focused pathology. On the other hand, ICIs, by blocking checkpoint molecules such as CTLA -4, PD -1, and PD -L1, disturb the equilibrium of immune tolerance, possibly triggering an unregulated T-cell attack on self-antigens.”

- Check if “rizhomelic” is the correct spelling.

- You presented some limitations that are external to your work (and inherent to the world of data). Heterogeneity is normal in retrospective studies. However, one real limitation, in my opinion, is the absence of initiatives to tackle heterogeneity in your work. For instance, while your choice to include studies irrespective of the diagnostic criteria increased your pool of studies, it compromised your internal validity.

- At the beginning of the manuscript, you highlighted the importance of PMR-mimics, but very little is dedicated to this topic throughout the text. Consider discussing it a bit more and providing the main mimics one should consider in daily practice. Also, consider providing an illustration or flowchart synthesizing the content of your research, aiming to help the clinician.

Comments on the Quality of English Language

Overall good. I understand only minor adjustments are necessary.

Author Response

please see the attachement

Reviewer 2 Report

Comments and Suggestions for Authors

It is an interesting topic.

Lines 43-45:  ,,Keywords: Polymyalgia Rheumatica; subset; Giant Cell Arteritis; Calcium Pyrophosphate Deposition disease; chondrocalcinosis; immune checkpoint inhibitor drugs; infection; vaccination; ultrasound.”

There are many keywords, a bit too many and I think there should be a maximum of 5.

Lines 141-143 : ,,The initial search yielded 2492 papers, of which 2389 articles were excluded based on title and abstract screening. A total of 103 articles underwent a full-length review, and  they were assessed for eligibility.”

 What is the search period for these papers?

Lines 163-166 : ,,However, all three studies agreed on the need to utilize imaging (US assessment, primarily) as well as measurement of other biomarkers in all patients who have clinical suspicion of PMR but not raised ESR and CRP, as already discussed and proposed in a  2018 editorial article [36]”

What are these biomarkers? Where did you write anything about them in the introduction subsection or the materials and methods subsection?

Lines 172-174: ,,Table 1. Polymyalgia Rheumatica with normal baseline acute-phase reactants. PMR = Polymyalgia Rheumatica, ACR/EULAR= European League 172 Against Rheumatism/American College of Rheumatology, MRI= magnetic resonance imaging, 18F-FDG PET-CT= 18F-labeled fluorodeoxyglucose-173 positron emission tomography-computed tomography, US= ultrasound, GC= glucocorticoids”

The title of a table is placed above it. Below the table is the Note, which may contain information about certain abbreviations in the table. I don't think a table is named both above and below.

Line 256 : ,,…Higher levels of inflammatory markers at onset [21, 56, 58, 61, 67, 70, 71], thrombocy…”

What inflammatory markers are you referring to?

It is good that you mentioned the strengths and limitations of the study.

This study may be the first to identify possible PRM clusters and I congratulate you for the idea.

The paper is less organized. This is also noticeable in the text, but most of all in the references where the reference number appears twice, some references have all the authors listed (ex 47,49,54,63,66,68,117,118), others only 2 authors and et al. (89,90). Then, in reference 97 the name of an author is listed incorrectly, and in number 4 the year is listed twice. There is no rule for including the year in the references, it appears immediately after the title of the paper, or somewhere towards the end. That is why it is difficult to follow each work from the references and very tiring to search for that work.

I appreciate the work done for this study.

My comments are only intended to make the paper better. Good luck!

Round 2

Reviewer 1 Report

Comments and Suggestions for Authors

Dear Authors,

Please find my replies below, after the "-->". Best of luck.

In “Half patients with initial PMR manifestations may experience a change in diagnosis at follow-up, often towards chronic arthritis,” consider checking the grammar.

As You suggested the grammar has been modified: “a change in final diagnosis has been experienced in about a half of patients with initial manifestations of PMR; in most cases their diagnosis was changed as chronic arthritis.”   (see lines 74-76) --> Ok.

- The part of “Definitions” in the Introduction is somewhat displaced in my opinion. In fact, I am not sure you actually applied your definitions throughout the study… If you believe these definitions are worthy to be mentioned, consider addressing them on the Discussion section.

We applied our definitions in the last column of all the tables, where all the studies have been checked for the possibility to define the reported subgroup as mimicking condition/other diagnosis. For this reason, we think that those definitions should be clarify at the beginning of the text. --> As I found this methodology hard to understand, other readers might find it also difficult. This is the problem with a large quantity of tables with a large quantity of information. Unlike what happens with the text—people usually navigate all the text—, most readers will most likely recur to tables to answer specific questions. I believe hardly ever someone will navigate line by line of the tables. But it is just my opinion. Nevertheless, consider clarifying this data will be provided in the tables, along with the criteria used to exclude every mimic. Also, please consider that if the patients included in the specific studies were not classified by any available PMR criteria, they would not fulfill your own definition of “PMR-mimicking conditions” (i.e.: “patients with clinical presentation of PMR fulfilling a validated set of diagnostic criteria (searching highest specificity) for PMR, but also fulfilling (within a short or long follow-up) a validated set of criteria for another nosology entity.”

- Considering your definitions, you have chosen to apply the Bird’s criteria because of sensitivity and the ACR/EULAR criteria because of specificity. But, instead of adding those, you decided to choose one of them. Considering your goal—to isolate the cases with the greatest certainty of true PMR— should we not use the combination of both—adding, thus, sensitivity and specificity? On the other hand, when searching for PMR-mimics, should we not use the most specific criteria (i.e.: reducing the false-positive)?

Thank You for the good question. From a statistic point of view, an initial screening of one disease should be made with a test (or criteria set) with the highest sensitivity, avoiding to lose false negatives. A successive search for specificity should be applied in a second moment, when we need to propose specific therapies or to propose cases for scientific purposes (clinical studies). As in PMR a specific diagnostic test is lacking, in the clinical practice the assignation of a diagnosis of PMR often rely on clinical judgement, diagnostic criteria (various sets) or classification criteria (ACR / EULAR 2012, however not produced for clinical purposes). For these motivations, we search for PMR subgroups in the various studies, independently from diagnosis modalities (the publication in peer reviewed journals should guarantee the quality of works). On the other hand, when we checked for PMR-mimics we should use criteria set with highest specificity. The error in the text has been corrected (see line 99). Thank You for the observations. --> The issue I highlighted refers not only to text corrections, but to the methodology itself. I apologize, but I do not believe it is enough to correct the word “specificity”. If you aim to exclude PMR-mimics based on comparing diagnostic cases with the highest specificity, only those studies should have been included.

- You stated that “Searches were performed regardless of language” but the exclusion criteria appear to have excluded non-English articles. So, you searched non-English articles only to exclude them later?

Thank You for the observation. The text has been corrected (non-english articles were excluded) (see line 120). --> Ok.

- Isn’t it curious that patients with normal APR have less systemic symptoms? Consider discussing this apparent paradox.

Systemic symptoms in PMR are related to increase of IL6 (and consequent increase of CRP, ESR, and anaemia). It is therefore plausible that patients with normal APR could have less systemic symptoms. All the scanty studies on this topic argued that PMR with normal APR could be an initial form of disease, with only incipient manifestations related to IL6 increase. --> I agree. Consider adding this information to the manuscript.

- How did Kara et al. included patients with PMR without elevated APR using the ACR criteria if elevated APR are required for entering the criteria?

In the study of Kara et al. the diagnosis of PMR in patients with normal APR was founded on imaging (US, MRI, PET) and clinical judgement. Moreover, a reasonably long follow-up (1 year) was adopted to exclude a different diagnosis. --> So, it would be imprecise to say that they used the ACR/EULAR criteria, right? Consider adjusting it in the table 1.

- In order to facilitate understanding for the readers and to allow them to judge the scientific soundness, could you provide the criteria Marsman et al. utilized in their study? Considering they are not validated, should they even be used?

In the study of Marsman only clinical judgement and clinical course (9 moths follow-up) were used to confirm diagnosis. Obviously, this modality of diagnosis and management cannot permit to diagnose associated diseases as asymptomatic CPPD or subclinical GCA. However, the publication on a respectable peer-reviewed journal guarantees the quality of the study. --> Unfortunately, I do not agree with you. Had we assumed that posture—blindly trusting a journal only because it is “respectable”—as scientifically solid and ethical throughout history, we would still not be vaccinating against measles due to fear of autism, simply because The Lancet published it. I believe that if you intend to objectively compare diagnoses, you must follow strict diagnostic/classification criteria; otherwise, your data simply cannot be trusted. I sincerely apologize if this response sounds harsh. I truly believe your work can contribute to the field, but I think the entire methodology must be revised first.

- This is just a detail, but temporal artery biopsy is not actually an imaging modality. Consider adjusting the column heading.

Thank You for the observation. We have modified the heading of all the tables (deleting “imaging” and adding “diagnostic tools”). --> Ok.

- While the tables provide a lot of content, I found it very tiresome to navigate all of them. Maybe you could consider another tool for showcasing those results (a graph or illustration, for instance). You can maintain the tables as supplementary for readers interested in more details.

You are right: the tables provide a lot of content. This is one of the strengths of our manuscript. We are sure you recognize the great work that underlies our manuscript. We think that this Your suggestion would completely change the essence of our review, without necessarily improving its quality and scientific soundness. Therefore, we decided not to accept it. We are sorry. --> Just a suggestion. No problem.

- In the studies within the ICI-PMR section, many reports rely on clinical judgement or no criteria whatsoever for diagnosing PMR. It is reasonable to use clinical judgement in daily clinical practice, but regarding scientific reporting, I understand that those articles defy the definition of “scientific”. Should we really be reporting them? Consider addressing this issue in your discussion.

We completely agree with You on this argument. Also, in the recent review of Hysa on ICI-PMR, the 2012 EULAR/ACR classification criteria for PMR were utilized in 12/24 observational studies (50%) and 9/14 case reports (64%), and the remaining studies relied only on physicians' clinical judgment for diagnosis, which is noteworthy given the broad spectrum of differential diagnoses in PMR. However, we choose to include all peer-reviewed papers on this topic independently from inclusion criteria. --> I have already discussed this before, but I understand that we would be better served by data comparing a handful of scientifically solid studies rather than dozens of studies with methodological issues.

- Consider checking the syntactic parallelism in: “a second with lesser systemic and articular inflammation, with persistence in PMR diagnosis and only GC.” We have modified some words in the paragraphs (see lines 348 and 354). --> Ok.

- Please cite the references supporting: “To date, primary PMR is considered a macrophage-focused pathology. On the other hand, ICIs, by blocking checkpoint molecules such as CTLA -4, PD -1, and PD -L1, disturb the equilibrium of immune tolerance, possibly triggering an unregulated T-cell attack on self-antigens.”

Done. Please, see the revised version of our manuscript (and ref. 109-110) --> Ok.

 - Check if “rizhomelic” is the correct spelling. Thank You for the observation. We have corrected the spelling (“rhizomelic) (see lines 454 and 458) --> Ok.

- You presented some limitations that are external to your work (and inherent to the world of data). Heterogeneity is normal in retrospective studies. However, one real limitation, in my opinion, is the absence of initiatives to tackle heterogeneity in your work. For instance, while your choice to include studies irrespective of the diagnostic criteria increased your pool of studies, it compromised your internal validity.

Your observation is correct, but we choose to analyse all the works where a subgroup was described, independently from the diagnostic modalities. As our work is the first describing these aspects of PMR, we think that all the peer-reviewed studies on this topic could be useful to have an initial idea of the heterogeneity of PMR, especially in diagnostic modalities. We reported these considerations in the revised version of our manuscript as final part within the Discussion section (lines 480-487). --> Already discussed. 

- At the beginning of the manuscript, you highlighted the importance of PMR-mimics, but very little is dedicated to this topic throughout the text. Consider discussing it a bit more and providing the main mimics one should consider in daily practice. Also, consider providing an illustration or flowchart synthesizing the content of your research, aiming to help the clinician.

As reported on the previous answer, the aim of this review was particularly focused on discovering subsets of real PMR, with lesser attention to the topic of PMR-mimics (this topic has been discussed in many other reviews and articles). --> Ok. But the diagnostic issue remains.

Author Response

See the attacchement
